# *Nodosilinea signiensis* sp. nov. (Leptolyngbyaceae, Synechococcales), a new terrestrial cyanobacterium isolated from mats collected on Signy Island, South Orkney Islands, Antarctica

**Ranina Radzi[1], Narongrit Muangmai[2], Paul Broady[3], Wan Maznah Wan Omar[1], Sebastien Lavoue[1], Peter Convey[4], Faradina Merican** [1] *

**1** School of Biological Sciences, Universiti Sains Malaysia, Minden, Penang, Malaysia, **2** Department of Fishery Biology, Faculty of Fisheries, Kasetsart University, Chatuchak, Bangkok, Thailand, **3** School of Biological Sciences, University of Canterbury, Christchurch, New Zealand, **4** British Antarctic Survey, Cambridge, United Kingdom

* faradina@usm.my

**Data Availability Statement:** All relevant data are within the paper. All new sequences generated in

## Abstract

Terrestrial cyanobacteria are very diverse and widely distributed in Antarctica, where they can form macroscopically visible biofilms on the surfaces of soils and rocks, and on benthic surfaces in fresh waters. We recently isolated several terrestrial cyanobacteria from soils collected on Signy Island, South Orkney Islands, Antarctica. Among them, we found a novel species of *Nodosilinea*, named here as *Nodosilinea signiensis* sp. nov. This new species is morphologically and genetically distinct from other described species. Morphological examination indicated that the new species is differentiated from others in the genus by cell size, cell shape, filament attenuation, sheath morphology and granulation. 16S rDNA phylogenetic analyses clearly confirmed that *N. signiensis* belongs to the genus *Nodosilinea*, but that it is genetically distinct from other known species of *Nodosilinea*. The D1–D1´ helix of the 16S–23S ITS region of the new species was also different from previously described *Nodosilinea* species. This is the first detailed characterization of a member of the genus *Nodosilinea* from Antarctica as well as being a newly described species.

## Introduction

Cyanobacteria are a widely distributed group of oxygenic photosynthetic prokaryotes that possess chlorophyll *a* and phycobiliproteins [1]. Despite their widespread occurrence and ecological importance, the taxonomy of cyanobacteria remains problematic. Cyanobacterial classification has recently undergone rapid revision based on the use of polyphasic approaches to define new taxa [2]. These approaches combine molecular characterization with cytomorphological and ecological characteristics and have been used to erect and describe new taxa and for the validation of classically described taxa [3, 4].

this study have been deposited in GenBank under accession numbers USMFM MN585774 and USMFM MN585775.

**Funding:** This study received funding support from YPASM Berth Support, YPASM Fellowship 304/PBIOLOGI/650963, RUI grant 1001/PBIOLOGI/811305 and Flagship grant 304/PBIOLOGI/650723/P131. Peter Convey is supported by NERC core funding to the BAS 'Biodiversity, Evolution and Adaptation' Team.

**Competing interests:** The authors have declared that no competing interests exist.

Amongst the different cyanobacteria groups, Leptolyngbyaceae (Synechococcales) is one of the most taxonomically challenging. Representatives of this family are morphologically simple, often with ambiguous and indistinct morphological features that lead to uncertainty in identification [5]. However, species within the group can be distinguished based on molecular evaluation. Ongoing taxonomic revisionary work to date has resulted in over 21 new genera being erected from the original genus *Leptolyngbya* [6–23]. The new erected genus were listed in various literature [6–11] and others are described [12–23].

The genus *Nodosilinea* [8] shows a clear phylogenetic separation from *Leptolyngbya* based on molecular assessment. The genus has been reported to possess a unique capability to form short convoluted lengths of trichomes within the bounding sheath, termed nodules. This was first documented in *L. nodulosa* [24]. The generic assignment of *L. nodolusa* was subsequently revised with the support of molecular phylogenetics and this species became the first member of the newly erected genus *Nodosilinea*, as *N. nodulosa* [24]. Subsequent studies have reported the ability to form nodules in all members of the genus when exposed to low-light conditions of $< 4$-$\mu$mol m$^{-2}$ s$^{-1}$ for 4 weeks. Some species have also been reported to perform nitrogen fixation [8, 24, 25]. Apart from nodule formation, the characteristics of the genus resemble those of *Leptolyngbya*, with uniseriate trichomes ranging from 0.5 to 3.5 μm wide within a thin sheath, similar vegetative and apical cell shapes, constriction at each cross wall and containing granules [26].

Representatives of the genus have been recorded from various habitats. *Nodosilinea* sp. LEGE 13457 and *Nodosilinea* sp. TM-3.1 were both isolated from the McMurdo Dry Valleys, Antarctica [27]. *N. ramsarensis* and *N. radiophila* were recorded from a thermal spring in Iran [28] and *N. nodulosa* from marine phytoplankton in the China Sea [24]. Other species are sub-aerial on walls (*N. epilithica* [8], *N. chupicuarensis* [25]) and occur in soils (*N. conica* [8]) and freshwater (*N. bijugata* [8, 29]).

In Antarctica, cyanobacteria are the most important primary colonizers of soils [30]. They play vital roles in soil stabilisation, photosynthetic carbon fixation, and the release of fixed nitrogen, whilst forming the base of the terrestrial food web [31]. Cyanobacteria are found across all geographical regions of Antarctica where they can form macroscopically visible mats, crusts or thin biofilms on the surfaces of soils and rocks and in streams, ponds and lakes, as well as occupying endolithic niches [32, 33].

*Leptolyngbya* sensu stricto is one of the most widely recorded genera in Antarctica [34–41]. However, detailed characterization combining morphological, ultrastructural and molecular approaches has been performed to date on only three named species (*L. bijugata* [42], *L. borchgrevinkii* and *L. frigida* [43]) and two unidentified species [44].

Three species of *Leptolyngbya* (i.e. *L. foveolarum*, *L. notata* and *L. perelegans*) were identified in a broad-scale floristic survey conducted on Signy Island, South Orkney Islands, in the 1970s [34]. Their taxonomic assignments, however, were based solely on morphological traits and have not subsequently been subjected to molecular assessment.

In this study, we report the polyphasic characterization of a cyanobacterial strain superficially resembling *Leptolyngbya*, recently isolated from soils collected on Signy Island.

## Materials and methods

### Ethics statement

New algal material was collected from Signy Island in the 2015/16 austral summer season under permit number 46/2015 issued under the United Kingdom Antarctic Act. The Department of Quarantine and Inspection Services Malaysia (MAQIS) provided import permit JPK1412016065849 to permit the samples to be imported to Malaysia (https://www.maqis.gov.

my/). No other specific permission was required for any other locations or activity included in this study. The locations are not privately owned or protected in any way. This study did not involve any endangered species or protected species.

## Sample origin and culture conditions

Mat samples were collected under permit as listed above from cracks and crevices in rocks and beneath loose fragments of stone on a west-facing slope below Robin Peak (60.6833˚ S, 45.6333˚ W) on Signy Island during the expedition of British Antarctic Survey in austral summer of 2015/16. All apparatus used for obtaining the samples was sterile and samples were stored and subsequently transported frozen (-20˚C) in sterile containers to the Universiti Sains Malaysia. Cyanobacterial strains were then brought into culture by inoculation on 1% agarised full strength BG-11 medium in Petri dishes [45]. Culture media were supplemented with 100 μg mL$^{-1}$ cycloheximide to prevent growth of eukaryotes [46]. Cultures were incubated at $15 \pm 2$˚C with 24 h light supplied by cool white fluorescent lamps at $< 4$ μmol m$^{-2}$ s$^{-1}$. After 4 weeks of incubation, algal growth developing on the plates was examined microscopically.

## Morphological characterization

Morphological examination was conducted using an Olympus BX-53 light microscope (Olympus America Inc., Center Valley, PA, USA) at 100 – 2000X magnification. Photomicrographs were taken. Illustrations were made with the aid of a *camera lucida*. Specimens were analysed based on morphological characteristics, particularly filament and trichome width, sheath morphology, cell colour, shape of intercalary and apical cells and presence of granules. Size measurements were made on 30 randomly chosen replicate specimens for each morphospecies. The characteristics of the strain studied were compared with descriptions in the literature [8, 25, 26].

## Transmission electron microscopy

Transmission electron microscopy (TEM) samples were fixed in McDowell-Trump fixative solution [47] prepared in 0.1 M phosphate buffer and later post-fixed with 1% osmium tetroxide. The fixed material was dehydrated in an ethanol series (50%, 75%, 95%, and 100%) and embedded in Spurr's resin [48] mixed in a rotator overnight. The ultrathin sections were treated with uranyl acetate and lead citrate to improve contrast [49]. Thin sections were collected on copper grids and were observed using a JEM 2000FX (JEOL, Tokyo, Japan) operating at 100 kV.

## Molecular analyses

DNA was extracted using the G-spin for bacteria genomic DNA extraction kit (iNtRON Biotechnology, Korea) following the manufacturer's protocol. DNA sample concentration was measured using a Thermo Scientific NanoDrop instrument. The 16S rDNA gene and the 16S–23S internal transcribed spacer (ITS) region were amplified using the polymerase chain reaction (PCR) and the combination of primers 2 (5'-GGG GGA TTT TCC GCA ATG GG-3') and 3 (5'-CGC TCT ACC AAC TGA GCT A- 3') for the 16S rRNA gene and primers 1 (5'- CTC TGT GTG CCT AGG TAT CC- 3') and 5 (5'-TGT AGC TCA GGT GGT TAG- 3') for the ITS region [50]. This resulted in products of approximately 1,600 bp for the 16S rRNA gene and 600 bp for the ITS region. The reaction mix comprised 2 μL of extracted DNA used in 50 μL reactions containing 1 μL of each forward and reverse primer, 21 μL of ultrapure water and 25 μL of *MyTaq™ Red Mix*, which is a pre-prepared

mixture of buffer, dNTPs and Taq polymerase (Bioline, United Kingdom). PCR was carried out using a Bio-Rad Thermal Cycler with standard parameters set as follows: 95˚C for 2 min, 95˚C for 15 sec, 55˚C for 15 sec (30 cycles), and 7 min 20 sec extension at 72˚C. Once the reaction was completed, the integrity of the PCR product was verified using a 2% agarose gel.

## Phylogenetic analyses and ITS folding

All sequences were edited and assembled using the Geneious 11.0 software package (Biomatters, http://www.geneious.com). Sequence alignments were prepared using the MUSCLE algorithm in Geneious 11.0 and then manually checked by eye. The dataset included 47 OTUs, consisting of sequences newly obtained in this study together with additional sequences retrieved from GenBank of closely related species of *Nodosilinea*, more distantly related species of *Leptolyngbya sensu stricto* and one outgroup taxon (*Gloeobacter violaceus* FR798924). Some *Leptolyngbya* sequence that were retrieved from GeneBank currently continue to be known as *Nodosilinea* and we identify these sequences with "*Leptolyngbya*" in the phylogenetic tree. All new sequences generated in this study have been deposited in GenBank under accession numbers USMFM MN585774 and USMFM MN585775.

Phylogenetic analyses were performed using two different methods: maximum likelihood (ML) and Bayesian inference (BI). Before ML and BI analyses, the best-fit model of DNA substitution was determined using the program Kakusan4 [51]. ML analyses were performed using with RaxML v7 [52] in Geneious 11.0 using the general time-reversible invariant-sites (GTRI) nucleotide substitution model with the default parameters. The bootstrap probability of each branch was calculated using 1000 replications. BI analyses were performed with the program MrBayes v3.1.2 [53]. Two independent analyses, each consisting of four Markov chains, were run simultaneously for 3,000,000 generations, sampling every 100 generations. Log likelihood and parameter values were assessed with Tracers ver. 1.5 [54]. A burn-in of 25% of saved trees was removed, and the remaining trees were used to calculate the Bayesian posterior probability values. ML and BI trees were edited with the program FigTree v1.3.1 [53].

The 16S–23S ITS region was used for modelling of secondary structure folding. The tRNA genes were identified using tRNAscan-SE 2 [55]. The secondary structure of the D1–D1´ helix was modelled using the Mfold WebServer with default conditions.

## Results

A single strain was successfully isolated from one sampling site and showed the diagnostic traits of the genus *Nodosilinea*. However, it has morphological characteristics that did not correspond to any previously described species. In addition, our phylogenetic study confirmed that this strain forms a distinct lineage within the genus with respect to the nucleotide sequences of both the 16S rDNA gene and the 16S-23S ITS region. We hereafter refer to this strain under the name *Nodosilinea signiensis* sp. nov.

*Class* Cyanophyceae
*Order* Synechococcales
*Family* Leptolyngbyaceae
Genus Nodosilinea
*Nodosilinea signiensis* sp. nov.

**Description**. (Culture conditions) Mat creeping on agar, pale blue-green to olive-green in colour. Filaments long, immotile, solitary, occasionally forming spirals under normal light conditions of 27 μmol m$^{-2}$ s$^{-1}$ (Fig 1A and 1C). Under low light intensity of <4 μmol m$^{-2}$ s$^{-1}$, uniseriate trichomes can lie parallel or twisted around one another within a common sheath

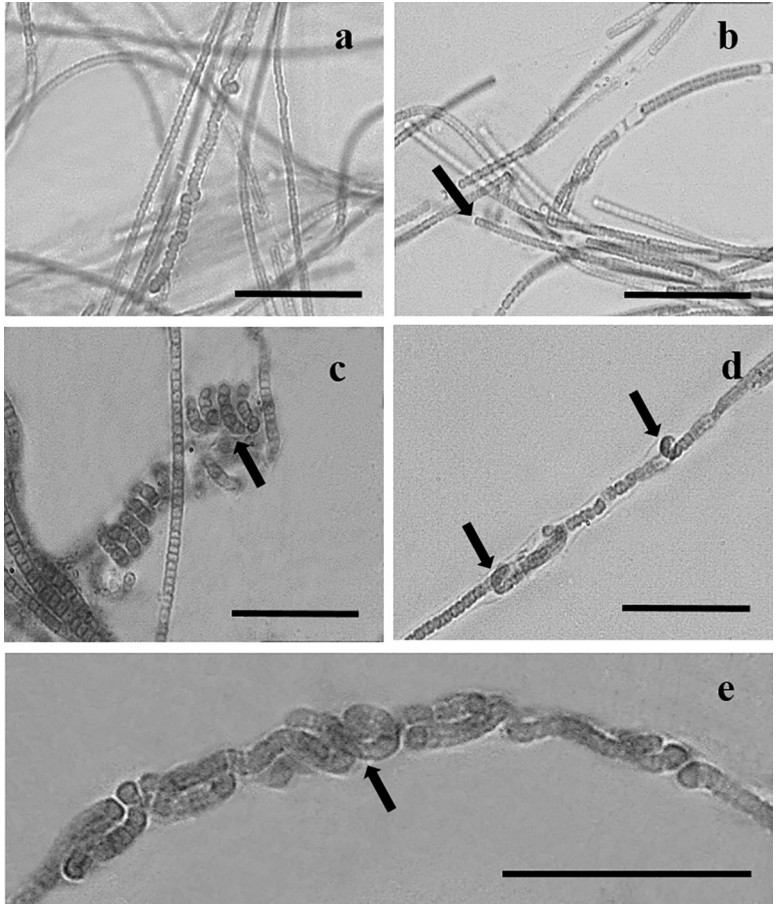

**Fig 1. *Nodosilinea signiensis* sp. nov. grown in culture. (A)**, filaments. **(B)**, rounded apical cell (arrow). **c**, spiral formation (arrow). **(D)** and **(E)**, trichomes twisted within the enclosing sheath, young trichome forming nodules (arrow). Scale bars: 20 μm.

(Fig 1D and 1E). This resembles nodule formation whereby filament width in these areas becomes wider, up to 5.0 μm wide (Fig 2C–2E). Cells 1.0 (1.5)– 2.0 μm wide, 1.0–2.0 (2.3) μm long, shorter to longer than wide, discoid to barrel-shaped (Fig 2A–2G). Apical cells rounded, non–capitate, without calyptra and lacking granules (Fig 2G). Sheath colourless, thin.

**Etymology.** *Nodosilinea signiensis*, Nodosilinea = "Knotted line" Perkeson et al. (2011); *signiensis* (sig.nie'n.sis) adj. signiensis = originated from Signy Island.

**Habitat.** Mats in cracks and crevices in rocks and beneath loose fragments of stone collected from west-facing slope below Robin Peak (60.6833˚ S, 45.6333˚ W).

**Occurrence.** South Orkney Islands, Signy Island.

**Observations.** In cultures incubated for 4 weeks under low light, filaments are entangled, curved or very occasionally form a spiral. Cell division without lengthening of the enclosing sheath causes the trichomes to sometimes bend and fragment to form two or three twisted structures that resemble nodules, although this was rarely observed.

**Morphological assessment.** The strain displays similar nodule formation to that of the seven currently described species within the genus. *Nodosilinea* sp. LEGE 13457 and *Nodosilinea* sp. TM-3.1, both originating from Antarctica [27], are not included here in the comparison as both strains lack detailed taxonomic characterization. Cell width of *Nodosilinea signiensis* sp. nov. falls within the range of all previously recorded species of *Nodosilinea*

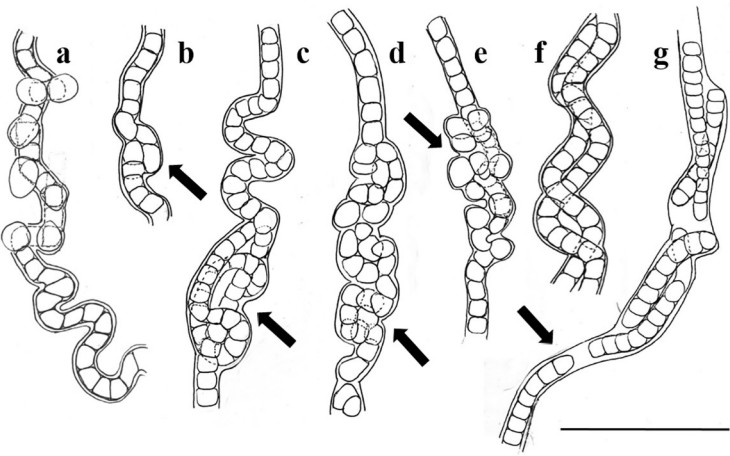

**Fig 2. Morphological characteristics of *Nodosilinea signiensis* sp. nov. (A)–(E)**, trichome variously curved, sometimes straight, tangled together or twisted, characteristic nodules (arrow). **(F) and (G)**, trichomes lying together within a common sheath, mature rounded apical cells (arrow). Scale bar: 10 μm.

(Table 1). Cell length closely resembles five of the seven described species (*N. conica*, *N. chupicuarensis*, *N. nodulosa*, *N. radiophila* and *N. ramsarensis*), while *N. epilithica* and *N. bijugata* have longer cells at 8 μm and 6.2 μm, respectively (Table 1). The vegetative cell shape of *N. signiensis* was distinct, being the only species showing a variety of cell shapes ranging from iso-diametric to longer than wide or shorter than wide. Cell constriction changed from slightly constricted to distinctly constricted as trichomes matured. In all other species of *Nodosilinea*, the cell shape was reported to be stable throughout the development of the trichome (Table 1). Trichomes of the present strain showed no attenuation towards the apex, in contrast to the abrupt tapering in *N. conica* (Table 1). *Nodosilinea signiensis* sp. nov. is the only species lacking cell granulation. Sheath development was very similar to *N. conica* and *N. chupicuarensis*, with the thin, colourless sheath occasionally becoming wide in mature filaments (Table 1). Three species, *N. conica*, *N. radiophila*, and *N. ramsarensis*, have been recorded from extreme environments (Table 1).

**Transmission electron microscopy.** Thylakoids were parietal (4–5 per cell), visible in longitudinal section but poorly seen under cross section (Fig 3). Nodule formation was not observed under TEM. Cyanophycin granules were visible among the thylakoids but carboxysomes were absent. The cell wall was simple, similar to *Leptolyngbya* species, and the sheath was distinct.

**Phylogenetic analysis.** The sequence dataset consisted of 1,482 bp, including gaps. Our tree suggests that "*Leptolyngbya antarctica*" should be reclassified into the genus *Nodosilinea*, as with several other sequence of "*Leptolyngbya*" deposited in GenBank. Partial sequences of the 16S rDNA of *N. signiensis* were distinct from other *Nodosilinea* sequences including "*Leptolyngbya antarctica*" by at least 2% ($\geq$ 29 differences). Both ML and BI analyses yielded an identical topology, and therefore only the ML tree with a $-ln$ L score of 11654.762 is presented (Fig 4). The 16S rDNA phylogeny clearly indicated that our strain was nested within the genus *Nodosilinea* (98% ML bootstrap percentage (BP) and 1.00 posterior probability (PP). The relationships within the *Nodosilinea* clade are complex and showed several low- to well-supported monophyletic groups. *Nodosilinea signiensis* forms a monophyletic group with two sequences of two specimens previously identified as "*Leptolyngbya*" sp. and one sequence of *Nodosilinea* sp. with high support from both ML (BP = 79%) and BI (PP = 93%).

**Table 1. Comparison of characteristics of the previously described eight species of *Nodosilinea* [8, 24, 25, 28] and *N. signiensis* sp. nov. ('?' indicates feature unknown).**

| Characters | *Nodosilinea signiensis* sp. nov. | *Nodosilinea epilithica* | *Nodosilinea bijugata* | *Nodosilinea conica* | *Nodosilinea chupicuarensis* | *Nodosilinea nodulosa* | *Nodosilinea radiophila* | *Nodosilinea ramsarensis* |
|---|---|---|---|---|---|---|---|---|
| **Cell length (μm)** | 1.0–2.0 (2.3) | 1.0–8.0 | 1.5–6.2 | 0.9–2.4 | 1.1–1.3 | 1.1–1.5 | 1.0–2.0 | (0.8) 1.0–1.5 |
| **Cell width (μm)** | 1.0 (1.5) | 1.5–2.5 | 1.5–1.7 | 2.5–2.7 | 1.2 | 1.2–2.4 | 2.0–5.0 | 1.0–2.0 |
| **Cell shape** | Isodiametric, longer than wide/ barrel shape | Barrel shaped, shorter to longer than wide | Isodiametric, longer than wide | Isodiametric, shorter than wide | Isodiametric | Isodiametric, longer than wide | Isodiametric, longer than wide | Isodiametric, longer than wide |
| **Cross-wall** | Slightly constricted to strongly constricted | Distinctly constricted | Slightly constricted | Slightly constricted | Constricted | Slightly constricted to strongly constricted | Distinctly constricted | Distinctly constricted |
| **Filaments** | Solitary, immotile, forming spiral | Forming nodules in low light | Rarely forming nodules | Rarely forming nodules | Multiseriate, motile, forming nodules | Forming nodules | No formation of nodules | Rarely forming nodules |
| **Attenuated** | Absent | ? | ? | Tapering abruptly | ? | ? | ? | ? |
| **Apical cells** | Rounded | Rounded | Rounded | Rounded | Dome-shaped | Rounded | ? | ? |
| **Granule** | Ungranulated | Granulated | Granulated | ? | Granulated at cross walls | Granulated at cross walls | Granulated at cross walls | Granulated |
| **Sheaths** | Very thin, colourless | Thin, colourless, occasionally becoming wide and diffluent | Often absent, thin, colourless | Soft, thin, colourless | Thin, clear | Thin, colourless, occasionally becoming wide and diffluent | Thin, colourless | Thin, colorless |
| **Special features** | Uniseriate trichome lie parallel or twisted around one another within a common sheath that resembles nodules | Cells typically barrel shaped after cell division, cylindrical in nondividing trichomes | Cells typically barrel shaped to spherical; inflated sheaths | Abundant nodule formation | Filament forming a tight spiral | ? | ? | ? |
| **Occurrence** | Soil—Signy Island, Antarctica | House wall -Peninsula Gargano, town of Vieste (Foggia), Italy. | Littoral zone– Eutrophic Lake Piaseczno, Poland | Sevilleta Long Term Ecological Research, New Mexico Soil -Chihuahuan Desert, USA | Stone monument surface—Central Mexico | Marine—South China Sea | Benthic mat in thermal spring (27 <C)—Talesh Mahalleh, Ramsar Iran | Soil around thermal spring (32 <C)— Khaksefid, Ramsar, Iran. |

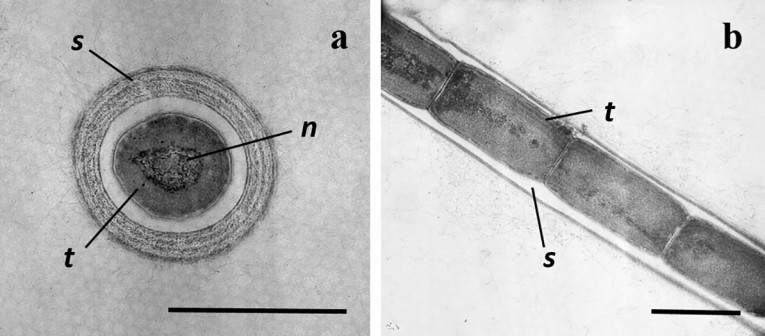

**Fig 3. TEM of *Nodosilinea signiensis*. (A)**, cross-section of a cell within its surrounding sheath. (B) longitudinal section of a trichome. Scale bar: 1 μm. *s* = sheath; *t* = thylakoids; *n* = nucleoplasm. Thylakoids are arranged more or less parallel in a parietal position.

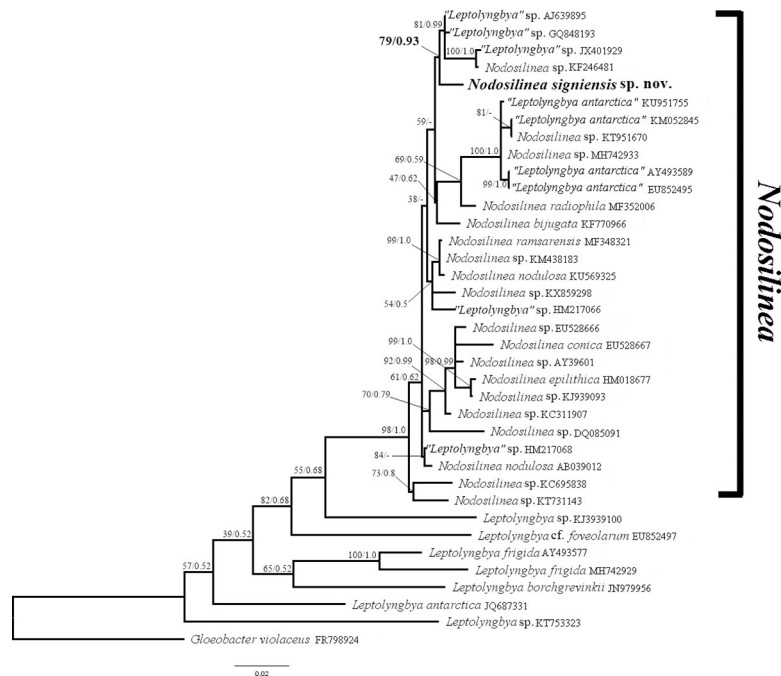

**Fig 4. 16S rRNA gene-based maximum likelihood phylogenetic tree showing the genetic distinctiveness and phylogenetic position of *Nodosilinea signiensis* sp. nov.** ML bootstrap values (left) followed by Bayesian posterior probabilities (right) on branches. Dashes indicate support values less than 50%. The sequence generated in this study is in boldface. *Gleobacter violaceous* FR798924 was the outgroup.

**ITS Secondary structure.** The D1–D1′ helix, a semi-conserved subregion of the 16S–23S ITS region, was examined in the genus *Nodosilinea*. The putative secondary structures of the D1–D1′ helix of *N. signiensis* sp. nov., together with those of other species of *Nodosilinea*, are presented in Fig 5. The D1–D1′ helix of eight *Nodosilinea* species contained 62–64 nucleotides, and seven of them (*N. signiensis* sp. nov, *N. radiophila* TMS2B, *N. ramsarensis* KH-S S2.6, *N. chupicuarensis* PCA471, *N. epilithica* Kovacik 1998/7, *N. nodulosa* UTEX 2910, *N. bijugata* Kovacik 1986/5a) possessed a 6 nucleotide unilateral bulge with highly conserved sequence (CACUCU), and shared the same basal stem structure (GACC–GGUC) (Fig 5). Despite their similar D1–D1′ helix structures, the sequences of *N. signiensis* sp. nov. differed from those of the seven previously described species.

## Discussion

Our study on a cyanobacterial strain superficially resembling *Leptolyngbya*, isolated from soils obtained on Signy Island, has resulted in the identification of a new species of *Nodosilinea*. *Nodosilinea signiensis* sp. nov. is differentiated from other previously described species based on cell size, cell shape, filament attenuation, sheath morphology, granulation and geographical distribution. Due to the simple morphology of this genus, some characteristics overlap between the species. However, the distinct cell shapes and the lack of filament attenuation separate *N. signiensis* sp. nov. from all previously described species. Cultivation clearly showed that nodule formation is only facilitated in low light conditions. Therefore, strains assigned to similar genera that have not been exposed to low light could have been misidentified. The occurrence of this species in an Antarctic terrestrial habitat further supports its separation from previously described species, based on the strikingly different biotope.

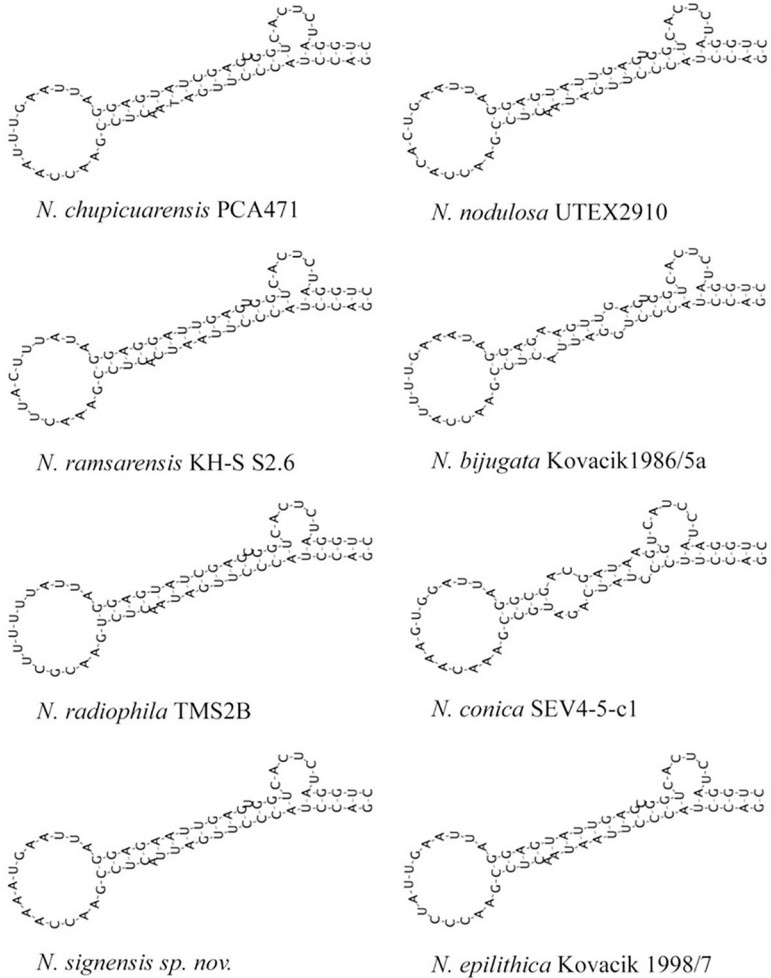

**Fig 5. Secondary structure based on 16S – 23S internal transcribed spacer (ITS) for eight species of *Nodosilinea*.**
*Nodosilinea signiensis* sp. nov. shared an identical unilateral bulge and basal stem structure with other species, except for *N. conica* SEV4-5-c1, but differed from other species in sequence detail.

Our genetic data further indicate that *N. signiensis* sp. nov. is distinct from other species of *Nodosilinea*. The 16S rDNA phylogenetic results are congruent with morphological examination in confirming that *N. signiensis* sp. nov. is a member of the genus *Nodosilinea*. Within *Nodosilinea*, *N. signiensis* sp. nov. showed similarity values of $< 98\%$ compared with species from the basal subclade. Stackbrandt and Goebel [56] recommended that the similarity cut-off for bacterial species recognition should be 97%. Recently, Stackbrandt and Ebers [57] raised the cut-off to 98.7%, a value later supported by Yarza et al. [58]. On this basis, *N. signiensis* sp. nov. represents a species distinct from the seven previously described *Nodosilinea* species. *Leptolyngbya antarctica* is recognised as the sister clade within the genus *Nodosilinea*. Morphological study of *L. antarctica* [59] and the genetic evidence confirm that *N. signiensis* sp. nov. and *L. antarctica* are clearly distinct from each other. The evolutionary relationships of *N. signiensis* sp. nov. within the genus remain largely unclear, possibly due to the limited number of variable sites in the 16S rDNA gene. Future use of multiple genetic makers is required to fully clarify phylogenetic relationships within *Nodosilinea*.

Modelling of the secondary structure of the ITS region indicated that the structure of the D1-D1'region of *N. signiensis* sp. nov. was closely similar to that of other species of *Nodosilinea*, even though the genetic sequences were distinctly different. Previous studies have shown that analysis of 16S rDNA gene phylogeny coupled with the secondary structure of the ITS region provides a suitable tool for separation of cyanobacteria species [8, 60, 61, 62, 63]. We note that the secondary structure of the ITS region in *Nodosilinea* is conserved among species. This study demonstrates that the integration of other lines of evidence, from morphological and genetic sequence data, provides an effective means to improve the taxonomy of species of *Nodosilinea*.

## Acknowledgments

We thank Dr Japareng Lalung for collection of the original soil sample on Signy Island. British Antarctic Survey staff at Signy research station are thanked for logistical and other practical support. This paper also contributes to the international SCAR 'State of the Antarctic Ecosystem' (AntEco) research programme. Peter Convey is supported by NERC core funding to the BAS 'Biodiversity, Evolution and Adaptation' Team.

## Author Contributions

**Conceptualization:** Ranina Radzi, Faradina Merican.

**Data curation:** Ranina Radzi, Narongrit Muangmai, Paul Broady, Peter Convey.

**Formal analysis:** Ranina Radzi, Narongrit Muangmai, Paul Broady, Peter Convey.

**Funding acquisition:** Faradina Merican.

**Investigation:** Ranina Radzi.

**Methodology:** Ranina Radzi, Paul Broady, Peter Convey, Faradina Merican.

**Resources:** Wan Maznah Wan Omar.

**Software:** Narongrit Muangmai.

**Supervision:** Paul Broady, Peter Convey, Faradina Merican.

**Validation:** Narongrit Muangmai, Peter Convey, Faradina Merican.

**Visualization:** Narongrit Muangmai, Paul Broady.

**Writing – original draft:** Ranina Radzi.

**Writing – review & editing:** Narongrit Muangmai, Paul Broady, Wan Maznah Wan Omar, Sebastien Lavoue, Peter Convey, Faradina Merican.

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
