## [Decision Letter · Decision Letter 0]

25 Jul 2019

PONE-D-19-15186

Nodosilinea signiensis sp. nov. (Leptolyngbyaceae, Synechococcales), a new terrestrial cyanobacterium isolated from soil collected on Signy Island, South Orkney Islands, Antarctica

PLOS ONE

Dear Dr Merican,

Thank you for submitting your manuscript to PLOS ONE. After careful consideration, we feel that it has merit but does not fully meet PLOS ONE’s publication criteria as it currently stands. Therefore, we invite you to submit a revised version of the manuscript that addresses the points raised during the review process.

We would appreciate receiving your revised manuscript by Sep 08 2019 11:59PM. To enhance the reproducibility of your results, we recommend that if applicable you deposit your laboratory protocols in protocols.io, where a protocol can be assigned its own identifier (DOI) such that it can be cited independently in the future. For instructions see: http://journals.plos.org/plosone/s/submission-guidelines#loc-laboratory-protocols

We look forward to receiving your revised manuscript.

Kind regards,

Susanna A Wood, PhD

Academic Editor

PLOS ONE

**Journal Requirements:**

**Additional Editor Comments (if provided):**

Thank you for you well written and conducted study. The manuscript has been evaluated by one reviewer and I have also carefully read the manuscript.

I commend you on the polyphasic approach and the careful photography and drawings. The study is quite limited in scope and it is probably better suited to an algal or taxonomic style journal. I have marked a few small corrections, suggestions and questions on the PDF and ask that you respond to each of these.

The reviewer has raised a number of major concerns related to single strain and limited corresponding environmental data. Please carefully consider these concerns and provide responses to each.

Thank you for submitting your work to PLoS One. I look forward to receiving you responses.

**Comments to the Author**

1. Is the manuscript technically sound, and do the data support the conclusions?

Reviewer #1: No

2. Has the statistical analysis been performed appropriately and rigorously? 

Reviewer #1: N/A

3. Have the authors made all data underlying the findings in their manuscript fully available?

Reviewer #1: No

4. Is the manuscript presented in an intelligible fashion and written in standard English?

Reviewer #1: Yes

5. Review Comments to the Author

Reviewer #1: Dear Authors,

The manuscript of Radzi et al.: “Nodosilinea signiensis sp. nov. (Leptolyngbyaceae, Synechococcales), a new terrestrial cyanobacterium isolated from soil collected on Signy Island, South Orkney Islands, Antarctica” introduce a new cyanobacterial species description: Nodosilinea signiensis. The manuscript is well written and well structured. However, the sampling size is not appropriate and does not support the conclusions of the study. I will not recommend this manuscript for publication as it is.

So far, the definition of bacterial ‘species’ is still debated. So, the combination of molecular and morphological characterization method is a good approach and a good choice. Nevertheless, my main concern about this manuscript is more about the material. Indeed, all the description of this ‘new’ taxa is based on the characterization of only one isolate in culture. This is not enough to define a new species under botanical or bacteriological codes. Also, the characterization of a new lineage requires the characterization of the organism in multiple environmental samples from diverse location, as well as the characterization of different strains isolated from diverse samples.

My second concern is about the fact that lineages belonging to Nodosilinea genus are very common in Antarctica. Thus, small non heterocytous filamentous cyanobacteria forms conspicuous mats in Antarctica and are well described in the literature (including strains) (eg. Taton et al 2006, Komarek et al 2007). So far, Letpolyngbya antarctica was reported as the most dominant taxa observed in mats. Recently, it was found that one of the clusters composing this polyphyletic taxon (firstly described by Taton et al. 2006) was the most abundant in saline lakes and was related to Nodosilinea (see Pessi et al. 2018). This taxon was also observed and strains were isolated from sample of sub Antarctic islands. They may be related to your isolate, and the latitude may partly explain their relatedness. For examples I know some strains from James Ross islands that are placed more or less in the same phylogenetic position. To my knowledge there are at least 5 strains available in collection culture which belong to this cluster from both maritime and continental Antarctic. I suggest you include the analysis of theses strains in your study.

I would like to encourage you to re submit in this journal or elsewhere a manuscript that will include detailed environmental data of samples, microscopic observation of environmental samples, more isolate from Signy islands as well as strains available in culture collections (BCCM/ULC; CCALA). Such a paper will be needed as there is still a lack of information about the OUT Leptolyngbya/ Nodosilinea which appeared to be one of the most dominant taxa present under these latitudes. It is also possible to look for location of all sequences that are similar by BLAST (ncbi).

Finally, there are few points for which I think the authors must not draw conclusions based on the data presented. Besides the fact that the sample size is not enough, differences of ITS sequences are not relevant here as D1 D1’ secondary structure is more relevant for this purpose. Genetic variation in ITS region can be very high in the same lineage it is observed in different taxa. Also, I think you can not emit conclusion on morphological characterization as presence of granulation as the other strains might not have been cultured with the same parameter and observed at a different time in their life cyles, as well as no conclusion could be made on the attenuation of the apex as it was not characterized in other strains.

Also, I regret that sequence was not available for reviewing purposes.

6. PLOS authors have the option to publish the peer review history of their article (what does this mean?). If published, this will include your full peer review and any attached files.

Reviewer #1: No

---

## [Author Response · Author response to Decision Letter 0]

11 Sep 2019

PLOSOne – revised manuscript by (11/9/2019)

Nodosilinea signiensis sp. nov. (Leptolyngbyaceae, Synechococcales), a new terrestrial cyanobacterium isolated from soil collected on Signy Island, South Orkney Islands, Antarctica

Response to reviewers’ comments

Reviewer’s comment 1: So far, the definition of bacterial ‘species’ is still debated. So, the combination of molecular and morphological characterization method is a good approach and a good choice. Nevertheless, my main concern about this manuscript is more about the material. Indeed, all the description of this ‘new’ taxa is based on the characterization of only one isolate in culture. This is not enough to define a new species under botanical or bacteriological codes. Also, the characterization of a new lineage requires the characterization of the organism in multiple environmental samples from diverse location, as well as the characterization of different strains isolated from diverse samples.

> Re: We thank the reviewer for the stimulating comment. Although theoretically we agree with the need to have more than one strain to describe the intraspecific variability, we would like to highlight three reasons that led us to describe this strain as new:

1) The significant morphological, ecological and molecular distinctiveness of this strain as compared to the other seven species previously described for the genus Nodosilinea. We believe that these three independent lines of evidence are a strong indication that our strain represents a new species.

2) The logistical difficulty to sample Signy Island makes it highly unlikely that we can return there in the near future, while the stochastic nature of field collections also provides no guarantee of any specific target taxon being present in any collections made. We wish to emphasize that we have surveyed 20 distinct collections in the initial examination of field samples, and this taxon was not present in any of these. However, upon establishing cultures of the field material from the 20 sites, we found the taxon growing well in plates inoculated only with the sample from Robin Peak. The restricted distribution (only occurring in 1 out of 20 sites) suggests that it is rare. Genetic data strongly supports the identity of the specimen being that a new species as proposed. 

3) We also think that in the face of the ongoing, rapid environmental change on Signy Island (warming climate, impact of enlarged fur seal population and on-going human visitors) that it is important to document and describe this distinctive species of Nodosilinea in the literature, for the use of other researchers. As it may be a species that could be threatened by these changes.

We hope that our study will serve as a reference for additional investigations on Antarctica cyanobacteria that might encounter similar strains elsewhere in the region.

Reviewer’s comment 2: My second concern is about the fact that lineages belonging to Nodosilinea genus are very common in Antarctica. Thus, small non heterocytous filamentous cyanobacteria forms conspicuous mats in Antarctica and are well described in the literature (including strains) (eg. Taton et al 2006, Komarek et al 2007). So far, Letpolyngbya antarctica was reported as the most dominant taxa observed in mats. Recently, it was found that one of the clusters composing this polyphyletic taxon (firstly described by Taton et al. 2006) was the most abundant in saline lakes and was related to Nodosilinea (see Pessi et al. 2018). This taxon was also observed and strains were isolated from sample of sub Antarctic islands. They may be related to your isolate, and the latitude may partly explain their relatedness. For examples I know some strains from James Ross islands that are placed more or less in the same phylogenetic position. To my knowledge there are at least 5 strains available in collection culture which belong to this cluster from both maritime and continental Antarctic. I suggest you include the analysis of theses strains in your study.

> Re: We fully agree with the reviewer that the comparison with Leptolyngbya antarctica is critical to determine whether our strain represents a new species or merely a population of L. antarctica. For that, we compared the 16S rRNA sequences of L. antarctica, which were provided by Taton et al. (2006) to describe this species. We convincingly demonstrate that our species is genetically distinct from L. antarctica (the similarity between the two species is below 98%). We agree with the reviewer that L. antarctica, as currently identified in the field, is a species complex and it is closely related to the genus Nodosilinea (and suggest that it should be transferred to this genus as Nodosilinea antarctica). The differences between the two were also supported by phenotypic evaluation of L. antarctica. The morphological characterization of L. antarctica as presented in Komarek (2007) showed no resemblance with our strain in having thinner and longer cells that are not constricted at the cross walls. The occurrence of nodules was also not reported for this species (Komarek, 2007).

We have included more sequences from thin and simple filamentous as suggested. 

Reviewer’s comment 3: I would like to encourage you to re submit in this journal or elsewhere a manuscript that will include detailed environmental data of samples, microscopic observation of environmental samples, more isolate from Signy islands as well as strains available in culture collections (BCCM/ULC; CCALA). Such a paper will be needed as there is still a lack of information about the OUT Leptolyngbya/ Nodosilinea which appeared to be one of the most dominant taxa present under these latitudes. It is also possible to look for location of all sequences that are similar by BLAST (ncbi).

>While we do agree with the value of this type of information, we have to emphasize that this is simply not possible, and has never been achieved in other ‘survey’ type studies, where multiple field samples are collected for laboratory processing at multiple specific locations and habitats differing widely in their micro environmental characteristics (see description and discussion in Convey et al. 2018 Polar Biology to emphasize this point). Similarly, this study did not (and could not) set out to make micro-environmental or microscopic descriptions of each of the many habitats sampled. 

Reviewer’s comment 4: Finally, there are few points for which I think the authors must not draw conclusions based on the data presented. Besides the fact that the sample size is not enough, differences of ITS sequences are not relevant here as D1 D1’ secondary structure is more relevant for this purpose. Genetic variation in ITS region can be very high in the same lineage it is observed in different taxa. 

> Re: We agree with the reviewer that differences in ITS sequences are not apparent in this study. However, this is a standard approach used in many studies (and failure to do so is itself often a source of reviewer criticism. For instance, see the studies of Perkerson et al. (2011) and Vázquez-Martínez et al. (2018), who used this region to distinguish within the Nodosilinea species. Comparison of the D1 D1’ made for all the species in the genus showed that seven out of eight previously species have identical secondary structure containing a unilateral bulge and basal stem, notwithstanding different sequences in every helix. Our findings are consistent with these previous studies (Perkerson et al. 2011, Vázquez-Martínez et al. 2018), which indicated the conserved feature of D1-D1’ structure in most members of the genus, including the newly proposed species N. signiensis

Also, I think you cannot emit conclusion on morphological characterization as presence of granulation as the other strains might not have been cultured with the same parameter and observed at a different time in their life cyles, as well as no conclusion could be made on the attenuation of the apex as it was not characterized in other strains. Also, I regret that sequence was not available for reviewing purposes.

> Re: Concerning the granulation, we only briefly mentioned the lack of granulation in the present specimen. We agree with the reviewer that this feature is not a diacritical characteristic unless there is a distinct pattern that is conserved under a range conditions in culture and at different ages in culture. Following normal practice, the sequences will be deposited in the GenBank should the manuscript be accepted for publication. 

We thank the reviewer for these critical comments.

---

## [Editor Report · Decision Letter 1]

14 Oct 2019

Nodosilinea signiensis sp. nov. (Leptolyngbyaceae, Synechococcales), a new terrestrial cyanobacterium isolated from soil collected on Signy Island, South Orkney Islands, Antarctica

PONE-D-19-15186R1

Dear Dr. Merican,

We are pleased to inform you that your manuscript has been judged scientifically suitable for publication and will be formally accepted for publication once it complies with all outstanding technical requirements.

With kind regards,

Susanna A Wood, PhD

Academic Editor

PLOS ONE
---

## [Editor Report · Acceptance letter]

23 Oct 2019

PONE-D-19-15186R1 

*Nodosilinea signiensis* sp. nov. (Leptolyngbyaceae, Synechococcales), a new terrestrial cyanobacterium isolated from mats collected on Signy Island, South Orkney Islands, Antarctica 

Dear Dr. Merican:

I am pleased to inform you that your manuscript has been deemed suitable for publication in PLOS ONE. Congratulations! Your manuscript is now with our production department. 

With kind regards,

on behalf of

Dr. Susanna A Wood 

Academic Editor

PLOS ONE